# OpenReview forum: "SafetyChat: Learning to Generate Physical Safety Warnings in Instructional Assistants"
_ICLR.cc/2026/Conference — Submitted to ICLR 2026_

### Official Review · Reviewer_Lqwh · 2025-10-23

**Soundness:** 2
**Presentation:** 3
**Contribution:** 1
**Rating:** 2
**Confidence:** 4

**Summary:**

The paper introduces SAFETYCHAT, a real-world, multi-turn benchmark for physical safety warnings in repair tasks and shows that post-training on this data materially improves LLMs’ ability to anticipate and communicate relevant hazards, surpassing several baselines and setting foundations for safer instructional assistants.

**Strengths:**

Novel dataset. Proposes a multi-turn benchmark focused on physical safety in automotive and electronics repair scenarios.

Clear task formulation. Formulate the problem as warning classification and warning generation, enabling systematic evaluation.

Convincing empirical evidence. Shows that popular LLMs underperform on these tasks, while simple post-training such as SFT and DPO yields substantial performance gains.

**Weaknesses:**

Beyond the two works cited, there are several recent datasets/benchmarks for physical-scenario safety in embodied/agent settings (e.g., SafeAgentBench; Agent-SafetyBench; robot constitutions/semantic safety; task-planning safety frameworks)[1,2,3,4]. Even if tasks differ, the paper should clarify what is new or harder here and why the proposed two tasks are distinctly more important than existing benchmarks.

The dataset is text-only, yet the source corpora (iFixit-Auto, wikiHow, TSB, iFixit-Elec) and many related benchmarks are multimodal (include images). For repair scenarios, users often rely on photos to communicate context. The paper should justify the text-only choice, discuss what is lost without images, and consider a multimodal extension.

Parts of the dataset are generated/evaluated with GPT-4o. To avoid model-specific bias, the pipeline should incorporate multiple diverse LLMs (and/or human adjudication) and report agreement and robustness across models.

Missing baselines likely to be strong. (1) RAG baseline: Given the availability of procedural documents (iFixit/wikiHow/TSB), a retrieval-augmented approach (retrieve instructions → summarize applicable warnings) is natural and may perform well; if no relevant instruction is found, classify as safe. (2) Reasoning models: Since the tasks require deciding whether to warn and what to warn about, chain-of-thought / reasoning models (or reasoning-enabled decoding) are appropriate baselines.

The post-training relies on standard SFT/DPO; there is no algorithmic contribution. Consider framing physical safety as hard/soft constraints during training or decoding (e.g., constrained optimization, safety filters, or control-theoretic constraints) to increase novelty and rigor.

Evaluation clarity issues. (1) Table 4 caption: It says “GPT-4-as-judge,” but the table has two columns: “GPT-4o Judge” and “Claude-3.7 Judge.” Please reconcile the caption with the content and clearly describe how GPT-4 (vs. GPT-4o) is used (and where it is first introduced). (2) Formatting: In §4.2, spacing differs between Query and Resp; ensure consistent English spaces throughout formulas and text.

[1]Yin, Sheng, et al. "Safeagentbench: A benchmark for safe task planning of embodied llm agents." arXiv preprint arXiv:2412.13178 (2024).
[2]Sermanet, Pierre, et al. "Generating robot constitutions & benchmarks for semantic safety." arXiv preprint arXiv:2503.08663 (2025). [3]Zhang, Zhexin, et al. "Agent-safetybench: Evaluating the safety of llm agents." arXiv preprint arXiv:2412.14470 (2024).
[4]Huang, Yuting, et al. "A Framework for Benchmarking and Aligning Task-Planning Safety in LLM-Based Embodied Agents." arXiv preprint arXiv:2504.14650 (2025).

**Questions:**

Did the authors encounter convergence or stability issues when training DPO given that preferred responses are human-authored while non-preferred responses are GPT-4o–generated (i.e., a clear source/distribution mismatch)? With a relatively small fine-tuning set, such shift can cause the model to learn source cues rather than preference signals and may hinder convergence.

---

> ### Author Response · Authors · 2025-12-01
> **Response to Lqwh**
>
> Thank you for your detailed review! We are glad that you found the paper proposes a “novel dataset” and “convincing empirical evidence”. We have addressed your comments below:
>
> > Beyond the two works cited, there are several recent datasets/benchmarks for physical-scenario safety in embodied/agent settings (e.g., SafeAgentBench; Agent-SafetyBench; robot constitutions/semantic safety; task-planning safety frameworks)[1,2,3,4]. Even if tasks differ, the paper should clarify what is new or harder here and why the proposed two tasks are distinctly more important than existing benchmarks.
>
> Thank you for sharing these papers! These papers all focus on the **safety of LLM agents**, which perform actions in an environment on behalf of a human. In this setting, the goal of these works is to prevent agents from harming humans, e.g., not crushing trash when a human is in the immediate proximity [2]. In contrast, our work focuses on the **safety of instructional LLMs**, which helps guide a human to complete a task through step-by-step instructions; here, humans take actions in the environment. In this setting, the goal is to warn the human about the potential physical dangers to them and other humans when taking the next step in the procedure. For instance, in the automotive repair domain, one of the warnings that could be issued to a human is to put on gloves before certain steps to help protect their hands from harmful fluids and also help them properly grip certain parts used in the step.
>
> Therefore, our task is related but distinct from prior work in LLM agent safety literature, as it focuses on safely guiding humans through tasks in the real world. However, we agree that a comparison of these works and this setting to our task would be valuable, and have added a small discussion on these papers in the Related Works section of the revised version of the paper.
>
>
>
> > The dataset is text-only, yet the source corpora (iFixit-Auto, wikiHow, TSB, iFixit-Elec) and many related benchmarks are multimodal (include images). For repair scenarios, users often rely on photos to communicate context. The paper should justify the text-only choice, discuss what is lost without images, and consider a multimodal extension.
>
> Thank you for pointing this out. We explored incorporating images alongside text in our experiments using state-of-the-art multi-modal methods. However, we did not find them particularly helpful. We hypothesize that the images provided in instructional documents are not crucial for generating safety warnings because oftentimes the images simply provide additional context for the textual instructions. However, we do not rule out the possibility of future work showing some way to use the images to improve warning generation performance. By including these images in the released dataset, we hope future research will further investigate how such visual demonstrations might enhance physical safety awareness. We have added this discussion to the Appendices.
>
> > Parts of the dataset are generated/evaluated with GPT-4o. To avoid model-specific bias, the pipeline should incorporate multiple diverse LLMs (and/or human adjudication) and report agreement and robustness across models.
>
> For the LLM-as-a-judge evaluation, we do use multiple LLMs (specifically GPT-4o and Claude 3.7) as the judge model and find that there is a high degree of agreement between their judgments as presented in Table 4. We also performed a calibration study of these judge models and found that the Pearson correlation between LLM judge ratings and human-provided ratings is 0.74, and the agreement rate is 90% (see line 410).
>
> We agree that it would be helpful to generate data with multiple LLMs. Since the paper submission, we have finished collecting conversations for an additional 161 procedures where a Qwen2.5-VL-32B-Instruct model was used as the assistant. We have updated the paper with a discussion of this data subset in X, and will release these conversations with the rest of the dataset upon publication.

---

> ### Author Response · Authors · 2025-12-01
> **Response to Lqwh (continued)**
>
> > Missing baselines likely to be strong. (1) RAG baseline: Given the availability of procedural documents (iFixit/wikiHow/TSB), a retrieval-augmented approach (retrieve instructions → summarize applicable warnings) is natural and may perform well; if no relevant instruction is found, classify as safe. (2) Reasoning models: Since the tasks require deciding whether to warn and what to warn about, chain-of-thought / reasoning models (or reasoning-enabled decoding) are appropriate baselines.
>
> We agree that it would be helpful to see the performance on additional baselines. We have accordingly added a retrieval augmented generation (RAG) baseline and a chain-of-thought (CoT) reasoning baseline in all of our classification result tables. The RAG baseline uses google search api for retrieval relevant information. The CoT baseline first considers when and when not to include each warning class. These baselines provide some small improvements on top of the direct prompting 0-shot baselines, although finetuning boosts the models even more. Please find the tables below.
>
> > The post-training relies on standard SFT/DPO; there is no algorithmic contribution. Consider framing physical safety as hard/soft constraints during training or decoding (e.g., constrained optimization, safety filters, or control-theoretic constraints) to increase novelty and rigor.
>
> We agree that the primary contribution of this paper is to introduce a novel task and dataset for physical safety. Therefore, we do not claim to make algorithmic contributions, but rather hope that our dataset enables such techniques in future work to further improve the physical safety warning capabilities of models. We have clarified this point in the introduction.
>
> > Evaluation clarity issues. (1) Table 4 caption: It says “GPT-4-as-judge,” but the table has two columns: “GPT-4o Judge” and “Claude-3.7 Judge.” Please reconcile the caption with the content and clearly describe how GPT-4 (vs. GPT-4o) is used (and where it is first introduced). (2) Formatting: In §4.2, spacing differs between Query and Resp; ensure consistent English spaces throughout formulas and text.
>
> Thank you for catching these formatting issues! We have fixed them in the revised version.

---

> > ### Author Response · Authors · 2025-12-02
> >
> > # Main classification table
> > | Warning Classes | BS-P | BS-R | SS-P | SS-R | CD-P | CD-R | JS-P | JS-R | WPE-P | WPE-R | Fo-P | Fo-R | Fl-P | Fl-R | Di-P | Di-R | All-P | All-R | All-F |
> > |-----------------|------|------|------|------|------|------|------|------|--------|--------|------|------|------|------|------|------|-------|-------|-------|
> > | Random          | 8    | 48   | 6    | 52   | 5    | 66   | 3    | 56   | 5      | 50     | 4    | 57   | 9    | 43   | 2    | 55   | 5     | 53    | 10    |
> > | No Warning      | 0    | 0    | 0    | 0    | 0    | 0    | 0    | 0    | 0      | 0      | 0    | 0    | 0    | 0    | 0    | 0    | 0     | 0     | 0     |
> > | Human           | 96   | 96   | 97   | 96   | 98   | 95   | 98   | 100  | 88     | 78     | 80   | 91   | 97   | 94   | 100  | 96   | 94    | 93    | 93    |
> > | GPT-4o-0-shot   | 57   | 79   | 41   | 17   | 69   | 38   | 60   | 83   | 20     | 44     | 100  | 7    | 46   | 26   | 70   | 64   | 58    | 45    | 44    |
> > | GPT-4o-8-shot   | 64   | 86   | 43   | 45   | 67   | 41   | 76   | 89   | 24     | 64     | 30   | 79   | 33   | 43   | 56   | 91   | 49    | 67    | 54    |
> > | Llama-3.1-8B-0-shot | 21 | 90 | 19 | 52 | 17 | 69 | 12 | 94 | 12 | 25 | 5 | 43 | 18 | 48 | 14 | 82 | 15 | 63 | 23 |
> > | Llama-3.1-8B-8-shot | 20 | 69 | 17 | 52 | 15 | 62 | 11 | 83 | 10 | 22 | 8 | 64 | 12 | 29 | 12 | 64 | 13 | 56 | 21 |
> > | Llama-3.1-8B-CoT-0-shot | 28 | 83 | 18 | 60 | 24 | 62 | 12 | 94 | 14 | 33 | 10 | 71 | 13 | 40 | 10 | 55 | 16 | 62 | 25 |
> > | Llama-3.1-8B-CoT-8-shot | 27 | 86 | 19 | 62 | 16 | 52 | 11 | 83 | 12 | 28 | 7 | 50 | 14 | 40 | 12 | 73 | 15 | 59 | 23 |
> > | Llama-3.1-8B-RAG-0-shot | 22 | 59 | 17 | 45 | 24 | 34 | 11 | 61 | 13 | 33 | 8 | 71 | 13 | 36 | 11 | 55 | 15 | 49 | 22 |
> > | Llama-3.1-8B-RAG-8-shot | 18 | 52 | 22 | 67 | 17 | 28 | 14 | 67 | 11 | 28 | 5 | 50 | 12 | 33 | 12 | 55 | 14 | 47 | 21 |
> > | **Llama-3.1-8B-SFT** | 59 | 45 | 89 | 40 | 87 | 45 | 94 | 89 | 80 | 33 | 55 | 43 | 71 | 29 | 100 | 73 | 79 | 50 | **60** |
> > | Qwen-2.5-7B-0-shot | 51 | 76 | 23 | 40 | 33 | 45 | 42 | 83 | 32 | 28 | 19 | 64 | 32 | 29 | 33 | 18 | 33 | 48 | 37 |
> > | Qwen-2.5-7B-8-shot | 60 | 86 | 24 | 40 | 37 | 45 | 38 | 83 | 31 | 22 | 18 | 50 | 33 | 31 | 60 | 27 | 38 | 48 | 39 |
> > | Qwen-2.5-7B-CoT-0-shot | 67 | 76 | 27 | 38 | 52 | 55 | 44 | 78 | 41 | 33 | 21 | 57 | 41 | 40 | 50 | 45 | 43 | 43 | 46 |
> > | Qwen-2.5-7B-CoT-8-shot | 66 | 72 | 28 | 40 | 48 | 48 | 48 | 89 | 38 | 31 | 19 | 50 | 39 | 31 | 43 | 27 | 41 | 49 | 43 |
> > | Qwen-2.5-7B-RAG-0-shot | 57 | 79 | 25 | 45 | 50 | 34 | 33 | 83 | 27 | 28 | 13 | 36 | 28 | 31 | 40 | 18 | 34 | 44 | 36 |
> > | Qwen-2.5-7B-RAG-8-shot | 57 | 83 | 20 | 40 | 30 | 28 | 41 | 89 | 28 | 25 | 13 | 43 | 19 | 21 | 30 | 27 | 30 | 45 | 34 |
> > | **Qwen-2.5-7B-SFT** | **71** | 34 | 81 | 52 | 69 | 62 | 94 | 94 | 72 | 36 | 78 | 50 | 62 | 24 | 78 | 64 | 76 | 52 | **60** |

---

> > > ### Author Response · Authors · 2025-12-02
> > > **Updated tables**
> > >
> > > Transfer learning experiments
> > >
> > > # Autorepair/Ford
> > >
> > > | Method - Binary F1        | SS   | CD   | JS   | ES   | UTC  | WPE  | MH   | RWD  | CE   | Di   | Average |
> > > |---------------------------|------|------|------|------|------|------|------|------|------|------|---------|
> > > | Random                    | 0.11 | 0.11 | 0.03 | 0.06 | 0.04 | 0.15 | 0.12 | 0.03 | 0.08 | 0.08 | 0.13 |
> > > | No Warning                | 0    | 0    | 0    | 0    | 0    | 0    | 0    | 0    | 0    | 0    | 0    |
> > > | GPT-4.1-mini-0s           | 0.11 | 0.19 | 0.11 | 0.40 | 0.02 | 0.16 | 0.16 | 0    | **0.32** | 0.37 | 0.18 |
> > > | GPT-4.1-mini-4s           | 0.30 | 0.62 | 0    | 0.33 | 0.04 | 0.21 | 0.30 | 0    | 0.27 | 0.48 | 0.26 |
> > > | Llama-3.1-8B-0s           | 0.16 | 0.22 | 0.11 | 0.08 | 0.05 | 0.28 | 0.03 | 0    | 0    | 0.30 | 0.12 |
> > > | Llama-3.1-8B-8s           | 0.19 | 0.22 | 0.10 | 0.15 | 0.05 | 0.27 | 0.14 | 0.18 | 0    | 0.36 | 0.17 |
> > > | Llama-3.1-8B-CoT-0s       | 0.30 | 0.28 | 0.05 | 0.10 | 0.04 | 0.19 | 0.08 | 0    | 0    | 0.32 | 0.14 |
> > > | Llama-3.1-8B-CoT-8s       | 0.27 | 0.31 | 0.10 | 0.10 | 0.07 | 0.31 | 0.16 | 0.07 | 0    | 0.32 | 0.17 |
> > > | Llama-3.1-8B-RAG-0s       | 0.16 | 0.22 | 0.11 | 0.08 | 0.05 | 0.28 | 0.03 | 0    | 0    | 0.30 | 0.12 |
> > > | Llama-3.1-8B-RAG-8s       | 0.19 | 0.22 | 0.10 | 0.15 | 0.05 | 0.27 | 0.14 | 0.18 | 0    | 0.36 | 0.17 |
> > > | **Llama-3.1-8B-SFT**      | **0.63** | **0.71** | 0.50 | 0.36 | **0.50** | **0.50** | **0.52** | **0.80** | 0.29 | 0.47 | **0.53** |
> > > | **Llama-3.1-8B-SFT(transferred from WikiHow)** | 0.63 | 0.57 | 0.40 | 0.36 | 0    | 0.57 | 0.29 | 0    | 0.10 | 0.52 | 0.34 |
> > > | Qwen-2.5-7B-0s            | 0.24 | 0.48 | 0    | 0.16 | 0.11 | 0.33 | 0.09 | 0.29 | 0    | 0.32 | 0.20 |
> > > | Qwen-2.5-7B-8s            | 0.24 | 0.48 | 0    | 0.24 | 0.08 | 0.22 | 0.17 | 0    | 0    | 0.25 | 0.17 |
> > > | Qwen-2.5-7B-CoT-0s        | 0.37 | 0.56 | 0    | 0.28 | 0.13 | 0.42 | 0.03 | 0    | 0    | 0.38 | 0.22 |
> > > | Qwen-2.5-7B-CoT-8s        | 0.35 | 0.53 | 0    | 0.26 | 0.06 | 0.40 | 0.25 | 0    | 0    | 0.32 | 0.22 |
> > > | Qwen-2.5-7B-RAG-0s        | 0.24 | 0.48 | 0    | 0.16 | 0.11 | 0.33 | 0.09 | 0.29 | 0    | 0.32 | 0.20 |
> > > | Qwen-2.5-7B-RAG-8s        | 0.24 | 0.48 | 0    | 0.24 | 0.08 | 0.22 | 0.17 | 0    | 0    | 0.25 | 0.17 |
> > > | **Qwen-2.5-7B-SFT**       | 0.62 | 0.65 | **0.55** | **0.37** | **0.50** | 0.47 | 0.37 | 0.67 | 0.18 | **0.50** | 0.49 |
> > >
> > > # Electronics/iFixit
> > > | Method - Binary F1        | ES   | HFS  | SWE  | WPE  | TU   | SS   | HMT  | Di   | Average |
> > > |---------------------------|------|------|------|------|------|------|------|------|---------|
> > > | Random                    | 0.22 | 0.02 | 0.06 | 0.05 | 0.08 | 0.03 | 0.02 | 0.12 | 0.07 |
> > > | No Warning                | 0    | 0    | 0    | 0    | 0    | 0    | 0    | 0    | 0    |
> > > | GPT-4o-0-shot             | 0.24 | 0.33 | 0.24 | 0.42 | **0.23** | 0.57 | 0 | 0.11 | 0.27 |
> > > | GPT-4o-8-shot             | 0.39 | 0.57 | 0.20 | 0.15 | 0.15 | 0.57 | 0 | 0.11 | 0.27 |
> > > | Llama-3.1-8B-0-shot       | 0.33 | 0.12 | 0    | 0.08 | 0.08 | 0.27 | 0 | 0 | 0.11 |
> > > | Llama-3.1-8B-8-shot       | 0.32 | 0.16 | 0.07 | 0.13 | 0.08 | 0.15 | 0 | 0 | 0.11 |
> > > | Llama-3.1-8B-CoT-0-shot   | 0.32 | 0.07 | 0.02 | 0.13 | 0.09 | 0.33 | 0 | 0.09 | 0.13 |
> > > | Llama-3.1-8B-CoT-8-shot   | 0.31 | 0.16 | 0.05 | 0.12 | 0.13 | 0.33 | 0 | **0.16** | 0.16 |
> > > | Llama-3.1-8B-RAG-0-shot   | 0.33 | 0.12 | 0    | 0.08 | 0.08 | 0.27 | 0 | 0 | 0.11 |
> > > | Llama-3.1-8B-RAG-8-shot   | 0.32 | 0.16 | 0.07 | 0.13 | 0.08 | 0.15 | 0 | 0 | 0.11 |
> > > | **Llama-3.1-8B-SFT**      | **0.56** | 0 | **0.31** | **0.67** | 0.15 | **0.80** | 0 | 0.11 | **0.32** |
> > > | **Llama-3.1-8B-SFT(Transferred from WikiHow)**| 0.36 | 0.50 | 0 | 0.50 | 0 | 0.67 | 0 | 0 | 0.25 |
> > > | Qwen-2.5-7B-0-shot        | 0.44 | **0.67** | 0.07 | 0.20 | 0.04 | 0.57 | 0 | 0 | 0.25 |
> > > | Qwen-2.5-7B-8-shot        | 0.43 | 0.57 | 0.16 | 0.22 | 0.12 | 0.67 | 0 | 0.11 | 0.29 |
> > > | Qwen-2.5-7B-CoT-0-shot    | 0.38 | **0.67** | 0.11 | 0.11 | 0.05 | 0.67 | 0 | 0.11 | 0.26 |
> > > | Qwen-2.5-7B-CoT-8-shot    | 0.38 | 0.33 | 0.08 | 0.09 | 0 | 0.57 | 0 | 0 | 0.18 |
> > > | Qwen-2.5-7B-RAG-0-shot    | 0.44 | **0.67** | 0.07 | 0.20 | 0.04 | 0.57 | 0 | 0 | 0.25 |
> > > | Qwen-2.5-7B-RAG-8-shot    | 0.43 | 0.57 | 0.16 | 0.22 | 0.12 | 0.67 | 0 | 0.11 | 0.29 |
> > > | **Qwen-2.5-7B-SFT**       | 0.52 | 0.50 | 0.11 | 0.44 | 0.22 | 0.67 | 0 | 0 | 0.31 |

---

### Official Review · Reviewer_DUDk · 2025-10-29

**Soundness:** 4
**Presentation:** 4
**Contribution:** 3
**Rating:** 6
**Confidence:** 3

**Summary:**

This paper introduces SafetyChat, a novel, multi-domain dataset designed to enhance Large Language Models (LLMs) in generating accurate and contextually relevant physical safety warnings for complex, multi-step instructional tasks like automotive and electronics repair. The authors argue that existing LLMs often fail to adequately address real-world physical hazards during conversational guidance, an oversight that becomes critical as these models integrate into voice assistants. SafetyChat consists of conversational benchmarks grounded in authentic repair procedures, with human annotators providing gold-standard safety rewrites for LLM responses that missed necessary warnings. Experiments confirm that while off-the-shelf LLMs perform poorly in hazard identification and warning generation, post-training on SafetyChat significantly improves their safety awareness, demonstrating a clear path toward developing safer instructional AI assistants.

**Strengths:**

- The paper is well written and well structured
- The authors present a good analysis of related work and datasets, as well as a good understanding of the state-of-the-art
- The paper introduces a potentially useful dataset of considerable size (6,391 annotated turns across 528 repair procedures) for evaluating AI assistants that provide instructions with greater safety awareness
- The SafetyChat dataset includes 1077 human-authored rewrites to address cases of missing warnings from GPT-4o responses.

**Weaknesses:**

- The experimental procedure could eventually be improved with some simple baseline approaches (e.g., see question 1). Essentially, since the main contribution is a benchmark dataset, demonstrating the performance of a more considerable number of base models and techniques would make up for a much more informative experiment.
- Given the nature of the contribution of the paper, having access to the dataset/anonymized code repository would be quite useful. For this reason, I can't comment on reproducibility.
- The experimental setting evaluates the models on a hold-out set. However, in this setting, out-of-distribution performance can be quite important (i.e., how do models perform on safety-sensitive tasks unrelated to any of the categories included in the train/validation subsets?). Something as simple as using one of the categories as a hold-out set could give readers an idea of how generalizable fine-tuning a model with this dataset could be (or employing something like a k-fold validation, where a fold would be a different category, for example).

**Questions:**

1. How does a fine-tuned model on this dataset perform (safety-wise) on out-of-distribution tasks, comparatively to its base version? Similarly, how would a prompting approach (i.e., using a model with specific instructions to be extra conscious on safety procedures, which I believe was the authors' approach with GPT-4o?) perform, comparatively to the fine-tuning and the base model approach, without such instructions? How would a model perform if, instead of fine-tuning, one employs retrieval-augmented generation instead (using the train set partition you defined)?
2. How many annotators were used? What was the selection criteria? In my opinion, this type of information should be better detailed (if not in the main body, at least in the appendix). I couldn't find much information on the annotation phase other than the interface developed.

---

> ### Author Response · Authors · 2025-12-01
> **Response to DUDk**
>
> Thank you for your helpful review! We are glad that you found the paper “paper introduces a potentially useful dataset of considerable size … for evaluating AI assistants that provide instructions with greater safety awareness”. We have addressed your comments below:
>
> > The experimental procedure could eventually be improved with some simple baseline approaches
>
> Thank you for pointing this out. We added more baseline results to the tables, including chain-of-thought prompting and retrieval augmented generation. These can be found in Tables 3, 6, 7, 8. In addition, we also add cross-domain knowledge transfer experiments, which can be found in these tables as well.
>
>
>
> > Given the nature of the contribution of the paper, having access to the dataset/anonymized code repository would be quite useful. For this reason, I can't comment on reproducibility.
>
> Thank you for sharing this concern. We created an anonymous git project and added it to the paper.
> https://anonymous.4open.science/r/SafetyChat-FAD6
>
> > The experimental setting evaluates the models on a hold-out set. However, in this setting, out-of-distribution performance can be quite important (i.e., how do models perform on safety-sensitive tasks unrelated to any of the categories included in the train/validation subsets?). Something as simple as using one of the categories as a hold-out set could give readers an idea of how generalizable fine-tuning a model with this dataset could be (or employing something like a k-fold validation, where a fold would be a different category, for example).
>
> Thank you for raising this point. We experimented with the cross-domain settings where the model is finetuned on the Autorepair/WikiHow Cars subsets and tested on Autorepair/Ford (5/10 unseen classes) and Electronics/iFixit subsets (6/8 unseen classes). We add a new result table to the appendix. The finetuned models perform significantly better than out-of-the box models on the new domains and unseen warning classes, which suggests that our finetuning is more than textual pattern matching. However, there are some other classes where the model struggles, potentially suggesting a negative transfer where the model over-predicts the unseen classes if their names are similar to seen classes, especially for closer domains. We leave it as a future work to explore more efficient generalization.
>
> Thank you for raising this concern. We test the model
>
> > How many annotators were used? What was the selection criteria? In my opinion, this type of information should be better detailed (if not in the main body, at least in the appendix). I couldn't find much information on the annotation phase other than the interface developed.
>
>  Our annotators were 5 undergraduate students who were rigorously trained following annotation guidelines that we developed and are partially presented in Appendix A2, A3, and A4. We also met with the annotators on a weekly basis to discuss ambiguous cases, and clarify the guidelines as needed to ensure high annotation quality. Annotators who consistently produced annotations that did not meet the high quality bar were not asked to annotate additional data, and their assigned data was reannotated. We agree these details are important, and have clarified them in Section 3 and the Appendix.

---

> > ### Author Response · Authors · 2025-12-02
> > **Updated tables**
> >
> > Below are transfer learning experiment results
> >
> > # Autorepair/Ford
> >
> > | Method - Binary F1        | SS   | CD   | JS   | ES   | UTC  | WPE  | MH   | RWD  | CE   | Di   | Average |
> > |---------------------------|------|------|------|------|------|------|------|------|------|------|---------|
> > | Random                    | 0.11 | 0.11 | 0.03 | 0.06 | 0.04 | 0.15 | 0.12 | 0.03 | 0.08 | 0.08 | 0.13 |
> > | No Warning                | 0    | 0    | 0    | 0    | 0    | 0    | 0    | 0    | 0    | 0    | 0    |
> > | GPT-4.1-mini-0s           | 0.11 | 0.19 | 0.11 | 0.40 | 0.02 | 0.16 | 0.16 | 0    | **0.32** | 0.37 | 0.18 |
> > | GPT-4.1-mini-4s           | 0.30 | 0.62 | 0    | 0.33 | 0.04 | 0.21 | 0.30 | 0    | 0.27 | 0.48 | 0.26 |
> > | Llama-3.1-8B-0s           | 0.16 | 0.22 | 0.11 | 0.08 | 0.05 | 0.28 | 0.03 | 0    | 0    | 0.30 | 0.12 |
> > | Llama-3.1-8B-8s           | 0.19 | 0.22 | 0.10 | 0.15 | 0.05 | 0.27 | 0.14 | 0.18 | 0    | 0.36 | 0.17 |
> > | Llama-3.1-8B-CoT-0s       | 0.30 | 0.28 | 0.05 | 0.10 | 0.04 | 0.19 | 0.08 | 0    | 0    | 0.32 | 0.14 |
> > | Llama-3.1-8B-CoT-8s       | 0.27 | 0.31 | 0.10 | 0.10 | 0.07 | 0.31 | 0.16 | 0.07 | 0    | 0.32 | 0.17 |
> > | Llama-3.1-8B-RAG-0s       | 0.16 | 0.22 | 0.11 | 0.08 | 0.05 | 0.28 | 0.03 | 0    | 0    | 0.30 | 0.12 |
> > | Llama-3.1-8B-RAG-8s       | 0.19 | 0.22 | 0.10 | 0.15 | 0.05 | 0.27 | 0.14 | 0.18 | 0    | 0.36 | 0.17 |
> > | **Llama-3.1-8B-SFT**      | **0.63** | **0.71** | 0.50 | 0.36 | **0.50** | **0.50** | **0.52** | **0.80** | 0.29 | 0.47 | **0.53** |
> > | **Llama-3.1-8B-SFT(transferred from WikiHow)** | 0.63 | 0.57 | 0.40 | 0.36 | 0    | 0.57 | 0.29 | 0    | 0.10 | 0.52 | 0.34 |
> > | Qwen-2.5-7B-0s            | 0.24 | 0.48 | 0    | 0.16 | 0.11 | 0.33 | 0.09 | 0.29 | 0    | 0.32 | 0.20 |
> > | Qwen-2.5-7B-8s            | 0.24 | 0.48 | 0    | 0.24 | 0.08 | 0.22 | 0.17 | 0    | 0    | 0.25 | 0.17 |
> > | Qwen-2.5-7B-CoT-0s        | 0.37 | 0.56 | 0    | 0.28 | 0.13 | 0.42 | 0.03 | 0    | 0    | 0.38 | 0.22 |
> > | Qwen-2.5-7B-CoT-8s        | 0.35 | 0.53 | 0    | 0.26 | 0.06 | 0.40 | 0.25 | 0    | 0    | 0.32 | 0.22 |
> > | Qwen-2.5-7B-RAG-0s        | 0.24 | 0.48 | 0    | 0.16 | 0.11 | 0.33 | 0.09 | 0.29 | 0    | 0.32 | 0.20 |
> > | Qwen-2.5-7B-RAG-8s        | 0.24 | 0.48 | 0    | 0.24 | 0.08 | 0.22 | 0.17 | 0    | 0    | 0.25 | 0.17 |
> > | **Qwen-2.5-7B-SFT**       | 0.62 | 0.65 | **0.55** | **0.37** | **0.50** | 0.47 | 0.37 | 0.67 | 0.18 | **0.50** | 0.49 |
> >
> > # Electronics/iFixit
> > | Method - Binary F1        | ES   | HFS  | SWE  | WPE  | TU   | SS   | HMT  | Di   | Average |
> > |---------------------------|------|------|------|------|------|------|------|------|---------|
> > | Random                    | 0.22 | 0.02 | 0.06 | 0.05 | 0.08 | 0.03 | 0.02 | 0.12 | 0.07 |
> > | No Warning                | 0    | 0    | 0    | 0    | 0    | 0    | 0    | 0    | 0    |
> > | GPT-4o-0-shot             | 0.24 | 0.33 | 0.24 | 0.42 | **0.23** | 0.57 | 0 | 0.11 | 0.27 |
> > | GPT-4o-8-shot             | 0.39 | 0.57 | 0.20 | 0.15 | 0.15 | 0.57 | 0 | 0.11 | 0.27 |
> > | Llama-3.1-8B-0-shot       | 0.33 | 0.12 | 0    | 0.08 | 0.08 | 0.27 | 0 | 0 | 0.11 |
> > | Llama-3.1-8B-8-shot       | 0.32 | 0.16 | 0.07 | 0.13 | 0.08 | 0.15 | 0 | 0 | 0.11 |
> > | Llama-3.1-8B-CoT-0-shot   | 0.32 | 0.07 | 0.02 | 0.13 | 0.09 | 0.33 | 0 | 0.09 | 0.13 |
> > | Llama-3.1-8B-CoT-8-shot   | 0.31 | 0.16 | 0.05 | 0.12 | 0.13 | 0.33 | 0 | **0.16** | 0.16 |
> > | Llama-3.1-8B-RAG-0-shot   | 0.33 | 0.12 | 0    | 0.08 | 0.08 | 0.27 | 0 | 0 | 0.11 |
> > | Llama-3.1-8B-RAG-8-shot   | 0.32 | 0.16 | 0.07 | 0.13 | 0.08 | 0.15 | 0 | 0 | 0.11 |
> > | **Llama-3.1-8B-SFT**      | **0.56** | 0 | **0.31** | **0.67** | 0.15 | **0.80** | 0 | 0.11 | **0.32** |
> > | **Llama-3.1-8B-SFT(Transferred from WikiHow)**| 0.36 | 0.50 | 0 | 0.50 | 0 | 0.67 | 0 | 0 | 0.25 |
> > | Qwen-2.5-7B-0-shot        | 0.44 | **0.67** | 0.07 | 0.20 | 0.04 | 0.57 | 0 | 0 | 0.25 |
> > | Qwen-2.5-7B-8-shot        | 0.43 | 0.57 | 0.16 | 0.22 | 0.12 | 0.67 | 0 | 0.11 | 0.29 |
> > | Qwen-2.5-7B-CoT-0-shot    | 0.38 | **0.67** | 0.11 | 0.11 | 0.05 | 0.67 | 0 | 0.11 | 0.26 |
> > | Qwen-2.5-7B-CoT-8-shot    | 0.38 | 0.33 | 0.08 | 0.09 | 0 | 0.57 | 0 | 0 | 0.18 |
> > | Qwen-2.5-7B-RAG-0-shot    | 0.44 | **0.67** | 0.07 | 0.20 | 0.04 | 0.57 | 0 | 0 | 0.25 |
> > | Qwen-2.5-7B-RAG-8-shot    | 0.43 | 0.57 | 0.16 | 0.22 | 0.12 | 0.67 | 0 | 0.11 | 0.29 |
> > | **Qwen-2.5-7B-SFT**       | 0.52 | 0.50 | 0.11 | 0.44 | 0.22 | 0.67 | 0 | 0 | 0.31 |

---

### Official Review · Reviewer_RX83 · 2025-10-31

**Soundness:** 3
**Presentation:** 2
**Contribution:** 2
**Rating:** 4
**Confidence:** 4

**Summary:**

The paper introduces SafetyChat, a multi-domain, multi-turn benchmark for teaching/evaluating LLMs to insert physical safety warnings into instructional dialogues, mainly for automotive and electronics repair. It also shows that small open models fine-tuned on this data can beat prompted GPT-4o on warning identification and on generating safety-aware rewrites.

**Strengths:**

- The paper introduces an interesting dataset in a space that is still underexplored - task-oriented, physical safety in instructional dialogues (automotive/electronics). This is a useful complement to the more common policy/content-safety benchmarks.

- Using iFixit/wikiHow/TSBs plus AR-style chat simulation keeps the dialogues realistic (images, long steps, workshop vs DIY differences). The Ford TSBs in particular justify the “high-stakes” framing; few safety datasets actually contain OEM technical bulletins.

- The task formulation is good: Separating (i) safety-warning identification from (ii) safety-aware response generation mirrors how production systems are actually built.

**Weaknesses:**

- Everything is repair-like: automotive, electronics, all from 3 sources. It’s plausible that the model is just learning “car-repair warning priors” (always tell them to park & cool down) and “electronics warning priors” (unplug, discharge capacitor) rather than contextual reasoning. The paper claims “realistic, multi-turn” but never shows cross-domain or out-of-domain transfer to kitchen, DIY home improvement. A cross-domain evaluation would make the “first step toward physically safe assistants” claim much stronger.

- The paper does not clearly describe the safety/technical expertise of the annotators (e.g., whether they had automotive/electrical background or were trained annotators following a rubric). Since the task is about physical risk, clarifying annotator qualification and quality control is important.

- On the modeling side, there is limited methodological novelty - mainly SFT and DPO on the collected data. This is fine for a dataset paper, but the work would be stronger with richer baselines (e.g. retrieval-augmented warning injection). The authors optionally can compare the approach of this paper: AURA: Affordance-Understanding and Risk-aware Alignment Technique for Large Language Models. https://arxiv.org/abs/2508.06124

- The evaluation relies heavily on LLM-as-a-judge. For a physical-safety benchmark, a small human study (even 15–20 domain-informed mechanics/DIYers/technicians) to validate that the model’s warnings are appropriate and non-redundant would significantly strengthen the empirical claims.

**Minor Comments**

- Image usage. Procedures often include images; but the generation task seems text-only. Say explicitly whether images are in the public release (some iFixit assets aren’t).

- A very intuitive baseline is “prepend domain-specific boilerplate chosen by IR from the taxonomy” (retrieve top-k warnings by BM25 over the step).

**Questions:**

See weaknesses

---

> ### Author Response · Authors · 2025-12-01
> **Response to RX83**
>
> Thank you for your thoughtful review! We are glad that you found that the paper “introduces an interesting dataset in a space that is still underexplored”. We have addressed your comments below:
>
> > Everything is repair-like ... The paper claims “realistic, multi-turn” but never shows cross-domain or out-of-domain transfer to kitchen, DIY home improvement. A cross-domain evaluation would make the “first step toward physically safe assistants” claim much stronger.
>
> Thank you for raising this point. We experimented with the cross-domain settings where the model is finetuned on the Autorepair/WikiHow Cars subsets and tested on Autorepair/Ford (5/10 unseen classes) and Electronics/iFixit subsets (6/8 unseen classes). We add a new result rows to Tables 7 and 8. The finetuned models perform significantly better than out-of-the box models on the new domains and unseen warning classes, which suggests that our finetuning is more than textual pattern matching. However, there are some other classes where the model struggles, potentially suggesting a negative transfer where the model over-predicts the unseen classes if their names are similar to seen classes, especially for closer domains. We leave it as a future work to explore more efficient generalization.
>
>
> > The paper does not clearly describe the safety/technical expertise of the annotators (e.g., whether they had automotive/electrical background or were trained annotators following a rubric). Since the task is about physical risk, clarifying annotator qualification and quality control is important.
>
> Due to the difficulty of recruiting expert annotators with relevant technical expertise, our annotators were 5 undergraduate students who were rigorously trained following annotation guidelines that we developed and are partially presented in Appendix A2, A3, and A4. The car repair guidelines, specifically, were also reviewed and approved by a member of a company that provides car repair services. We also met with the annotators on a weekly basis to discuss ambiguous cases, and clarify the guidelines as needed to ensure high annotation quality. Annotators who consistently produced annotations that did not meet the high-quality bar were not asked to annotate additional data, and their assigned data was reannotated. We have clarified these details in Section 3 and the Appendix.
>
> > On the modeling side, there is limited methodological novelty - mainly SFT and DPO on the collected data. This is fine for a dataset paper, but the work would be stronger with richer baselines (e.g. retrieval-augmented warning injection). The authors optionally can compare … [to] AURA
>
> We agree that it would be helpful to see the performance on additional baselines. We have accordingly added a retrieval augmented generation (RAG) baseline in all of our result tables. The RAG baseline provides some incremental improvements compared to 0-shot baselines, although finetuning more substantially improves performance.
>
> Thank you for bringing our attention to the AURA paper. Because our dataset does not have reasoning step-level annotations, we are unable to train a PRM like AURA. However, we agree this work is highly relevant and have added a citation and short discussion of this paper in the Related Works section (line 98).
>
> > The evaluation relies heavily on LLM-as-a-judge. For a physical-safety benchmark, a small human study (even 15–20 domain-informed mechanics/DIYers/technicians) to validate that the model’s warnings are appropriate and non-redundant would significantly strengthen the empirical claims.
>
> We agree that calibration of the LLM judge with humans is critical for faithful evaluation. To measure this, we performed a calibration study, finding that the Pearson correlation between LLM judge ratings and human-provided ratings is 0.74 and the agreement rate is 90% (see line 410). Additionally, from Table 4, the human-oracle rewritten responses are almost always preferred to the original GPT-4o response, indicating the GPT-4o judge is not simply preferring its own response. Finally, even the best model after fine-tuning only achieves a roughly 50% win-rate against the human-oracle, indicating that the fine-tuned model responses are nearly indistinguishable from the oracle responses, which is the upper-bound of possible improvement. We have highlighted these points in Section 5.3 of the revised paper.
>
> > Image usage. Procedures often include images; but the generation task seems text-only. Say explicitly whether images are in the public release (some iFixit assets aren’t)
>
> Following the MyFixit paper, our public release will contain direct links to all images used in the procedures. Assets in the Ford split will also be accessible via a direct link.

---

> > ### Author Response · Authors · 2025-12-01
> > **Response to RX83 (continued)**
> >
> > > A very intuitive baseline is “prepend domain-specific boilerplate chosen by IR from the taxonomy” (retrieve top-k warnings by BM25 over the step).
> >
> > Thank you for suggesting this approach. We tried this BM25(top 3 classes) + generation baseline, but found no significant difference between it and the current baseline. We assume it is because the current prompt has a short and complete list of taxonomy definitions, such that it works sufficiently as the context needed by the LLM. In addition, we also tried another RAG baseline where the model seeks additional external knowledge from the web using the Google Search api. These new results can be found in Tables 3, 6, 7, and 8. We find that this baseline does provide some small improvements on top of the direct prompting 0-shot baselines, although finetuning boosts the models even more.

---

> > ### Author Response · Authors · 2025-12-02
> > **Updated tables**
> >
> > Transfer learning experiments
> >
> > # Autorepair/Ford
> >
> > | Method - Binary F1        | SS   | CD   | JS   | ES   | UTC  | WPE  | MH   | RWD  | CE   | Di   | Average |
> > |---------------------------|------|------|------|------|------|------|------|------|------|------|---------|
> > | Random                    | 0.11 | 0.11 | 0.03 | 0.06 | 0.04 | 0.15 | 0.12 | 0.03 | 0.08 | 0.08 | 0.13 |
> > | No Warning                | 0    | 0    | 0    | 0    | 0    | 0    | 0    | 0    | 0    | 0    | 0    |
> > | GPT-4.1-mini-0s           | 0.11 | 0.19 | 0.11 | 0.40 | 0.02 | 0.16 | 0.16 | 0    | **0.32** | 0.37 | 0.18 |
> > | GPT-4.1-mini-4s           | 0.30 | 0.62 | 0    | 0.33 | 0.04 | 0.21 | 0.30 | 0    | 0.27 | 0.48 | 0.26 |
> > | Llama-3.1-8B-0s           | 0.16 | 0.22 | 0.11 | 0.08 | 0.05 | 0.28 | 0.03 | 0    | 0    | 0.30 | 0.12 |
> > | Llama-3.1-8B-8s           | 0.19 | 0.22 | 0.10 | 0.15 | 0.05 | 0.27 | 0.14 | 0.18 | 0    | 0.36 | 0.17 |
> > | Llama-3.1-8B-CoT-0s       | 0.30 | 0.28 | 0.05 | 0.10 | 0.04 | 0.19 | 0.08 | 0    | 0    | 0.32 | 0.14 |
> > | Llama-3.1-8B-CoT-8s       | 0.27 | 0.31 | 0.10 | 0.10 | 0.07 | 0.31 | 0.16 | 0.07 | 0    | 0.32 | 0.17 |
> > | Llama-3.1-8B-RAG-0s       | 0.16 | 0.22 | 0.11 | 0.08 | 0.05 | 0.28 | 0.03 | 0    | 0    | 0.30 | 0.12 |
> > | Llama-3.1-8B-RAG-8s       | 0.19 | 0.22 | 0.10 | 0.15 | 0.05 | 0.27 | 0.14 | 0.18 | 0    | 0.36 | 0.17 |
> > | **Llama-3.1-8B-SFT**      | **0.63** | **0.71** | 0.50 | 0.36 | **0.50** | **0.50** | **0.52** | **0.80** | 0.29 | 0.47 | **0.53** |
> > | **Llama-3.1-8B-SFT(transferred from WikiHow)** | 0.63 | 0.57 | 0.40 | 0.36 | 0    | 0.57 | 0.29 | 0    | 0.10 | 0.52 | 0.34 |
> > | Qwen-2.5-7B-0s            | 0.24 | 0.48 | 0    | 0.16 | 0.11 | 0.33 | 0.09 | 0.29 | 0    | 0.32 | 0.20 |
> > | Qwen-2.5-7B-8s            | 0.24 | 0.48 | 0    | 0.24 | 0.08 | 0.22 | 0.17 | 0    | 0    | 0.25 | 0.17 |
> > | Qwen-2.5-7B-CoT-0s        | 0.37 | 0.56 | 0    | 0.28 | 0.13 | 0.42 | 0.03 | 0    | 0    | 0.38 | 0.22 |
> > | Qwen-2.5-7B-CoT-8s        | 0.35 | 0.53 | 0    | 0.26 | 0.06 | 0.40 | 0.25 | 0    | 0    | 0.32 | 0.22 |
> > | Qwen-2.5-7B-RAG-0s        | 0.24 | 0.48 | 0    | 0.16 | 0.11 | 0.33 | 0.09 | 0.29 | 0    | 0.32 | 0.20 |
> > | Qwen-2.5-7B-RAG-8s        | 0.24 | 0.48 | 0    | 0.24 | 0.08 | 0.22 | 0.17 | 0    | 0    | 0.25 | 0.17 |
> > | **Qwen-2.5-7B-SFT**       | 0.62 | 0.65 | **0.55** | **0.37** | **0.50** | 0.47 | 0.37 | 0.67 | 0.18 | **0.50** | 0.49 |
> >
> > # Electronics/iFixit
> > | Method - Binary F1        | ES   | HFS  | SWE  | WPE  | TU   | SS   | HMT  | Di   | Average |
> > |---------------------------|------|------|------|------|------|------|------|------|---------|
> > | Random                    | 0.22 | 0.02 | 0.06 | 0.05 | 0.08 | 0.03 | 0.02 | 0.12 | 0.07 |
> > | No Warning                | 0    | 0    | 0    | 0    | 0    | 0    | 0    | 0    | 0    |
> > | GPT-4o-0-shot             | 0.24 | 0.33 | 0.24 | 0.42 | **0.23** | 0.57 | 0 | 0.11 | 0.27 |
> > | GPT-4o-8-shot             | 0.39 | 0.57 | 0.20 | 0.15 | 0.15 | 0.57 | 0 | 0.11 | 0.27 |
> > | Llama-3.1-8B-0-shot       | 0.33 | 0.12 | 0    | 0.08 | 0.08 | 0.27 | 0 | 0 | 0.11 |
> > | Llama-3.1-8B-8-shot       | 0.32 | 0.16 | 0.07 | 0.13 | 0.08 | 0.15 | 0 | 0 | 0.11 |
> > | Llama-3.1-8B-CoT-0-shot   | 0.32 | 0.07 | 0.02 | 0.13 | 0.09 | 0.33 | 0 | 0.09 | 0.13 |
> > | Llama-3.1-8B-CoT-8-shot   | 0.31 | 0.16 | 0.05 | 0.12 | 0.13 | 0.33 | 0 | **0.16** | 0.16 |
> > | Llama-3.1-8B-RAG-0-shot   | 0.33 | 0.12 | 0    | 0.08 | 0.08 | 0.27 | 0 | 0 | 0.11 |
> > | Llama-3.1-8B-RAG-8-shot   | 0.32 | 0.16 | 0.07 | 0.13 | 0.08 | 0.15 | 0 | 0 | 0.11 |
> > | **Llama-3.1-8B-SFT**      | **0.56** | 0 | **0.31** | **0.67** | 0.15 | **0.80** | 0 | 0.11 | **0.32** |
> > | **Llama-3.1-8B-SFT(Transferred from WikiHow)**| 0.36 | 0.50 | 0 | 0.50 | 0 | 0.67 | 0 | 0 | 0.25 |
> > | Qwen-2.5-7B-0-shot        | 0.44 | **0.67** | 0.07 | 0.20 | 0.04 | 0.57 | 0 | 0 | 0.25 |
> > | Qwen-2.5-7B-8-shot        | 0.43 | 0.57 | 0.16 | 0.22 | 0.12 | 0.67 | 0 | 0.11 | 0.29 |
> > | Qwen-2.5-7B-CoT-0-shot    | 0.38 | **0.67** | 0.11 | 0.11 | 0.05 | 0.67 | 0 | 0.11 | 0.26 |
> > | Qwen-2.5-7B-CoT-8-shot    | 0.38 | 0.33 | 0.08 | 0.09 | 0 | 0.57 | 0 | 0 | 0.18 |
> > | Qwen-2.5-7B-RAG-0-shot    | 0.44 | **0.67** | 0.07 | 0.20 | 0.04 | 0.57 | 0 | 0 | 0.25 |
> > | Qwen-2.5-7B-RAG-8-shot    | 0.43 | 0.57 | 0.16 | 0.22 | 0.12 | 0.67 | 0 | 0.11 | 0.29 |
> > | **Qwen-2.5-7B-SFT**       | 0.52 | 0.50 | 0.11 | 0.44 | 0.22 | 0.67 | 0 | 0 | 0.31 |

---

### Official Review · Reviewer_6je4 · 2025-10-31

**Soundness:** 2
**Presentation:** 2
**Contribution:** 2
**Rating:** 4
**Confidence:** 3

**Summary:**

This paper addresses an important and overlooked problem: the inability of Large Language Models (LLMs) to generate context-aware safety warnings when acting as instructional assistants for complex tasks with physical risks (e.g., automotive or electronics repair). The authors point out that even advanced models like GPT-4o often fail to provide critical physical safety instructions.

To tackle this issue, the paper makes two main contributions:
1.  **The SAFETYCHAT Dataset**: A novel, multi-domain (automotive and electronics repair) large-scale conversational dataset. It is built upon real-world repair guides (such as iFixit, wikiHow, and TSBs) and collected through multi-turn, role-playing dialogues between human annotators and GPT-4o.
2.  **Safety Alignment Experiments**: The authors performed Supervised Finetuning (SFT) and Direct Preference Optimization (DPO) on open-source models (e.g., Llama-3.1-8B) using SAFETYCHAT.

Experimental results show that models trained on SAFETYCHAT significantly outperform (and even surpass) GPT-4o on tasks involving the classification and generation of physical safety warnings. This demonstrates that alignment with high-quality, domain-specific data can effectively enhance the physical safety awareness of LLMs.

**Strengths:**

1.  **Importance and Novelty of the Problem**: The paper addresses a critical and under-researched area: the **physical safety** of LLMs. As models are increasingly integrated into smart glasses, AR/VR, or embodied agents, the ability to foresee and warn against physical hazards during instructional tasks is paramount.
2.  **High-Quality Dataset Construction**: The SAFETYCHAT dataset is a core contribution of this work. It is built on authoritative, real-world repair guides (including professional TSBs) and employs a rigorous collection methodology. Notably, having human annotators **rewrite** GPT-4o's responses that missed safety warnings provides an exceptionally high-quality training signal for SFT and DPO.
3.  **Effective Alignment Strategy**: The experiments demonstrate that alignment using SFT and DPO on a domain-specific dataset is extremely effective. It is noteworthy that the finetuned 8B model surpasses GPT-4o on physical safety tasks, suggesting that for specific safety concerns, targeted data and alignment are more effective than larger, general-purpose models.

**Weaknesses:**

1.  **The Inherent Paradox of LLM-as-a-Judge Evaluation**: A fundamental weakness lies in the evaluation methodology. The paper first establishes that GPT-4o is deficient in identifying physical safety hazards, yet it paradoxically relies on this same "incapable" model as the primary "judge" for evaluating the safety generation tasks. This contradiction undermines the validity of the results, as a small-scale human verification is insufficient to resolve the concern that the judge model has the very blind spots it is supposed to be evaluating.

2.  **Lack of a Multimodal Evaluation**: Despite introductory scenarios like "smart glasses" and the use of images during data collection, all experiments remain purely textual. The evaluation framework fails to assess the model's ability to perceive physical danger from visual context, which is a critical component of the very problem the paper aims to solve.

3.  **The Dataset Name "SafetyChat" is a Significant Overclaim**: The name implies a general-purpose safety model, whereas the work is narrowly focused only on procedural physical safety for automotive and electronics repair. This is misleading and potentially dangerous, as it falsely suggests the model is suitable for other safety domains (like social, privacy, or even medical safety) where it has no training.

**Questions:**

How do the authors view the generalization capabilities of models trained on SAFETYCHAT? For instance, could a model trained on automotive repair data handle a physical safety warning for "bicycle repair," which it has never seen? Has the model learned specific textual patterns for "jack safety" and "battery safety," or has it grasped a more general concept of "physical harm avoidance"?

---

> ### Author Response · Authors · 2025-12-01
> **Response to 6je4**
>
> Thank you for your detailed review! We are glad that you found the paper “addresses a critical under-researched area”. We have addressed your comments below:
>
> >   The paper first establishes that GPT-4o is deficient in identifying physical safety hazards, yet it paradoxically relies on this same "incapable" model as the primary "judge" for evaluating the safety generation tasks …
>
> To perform the LLM-as-a-judge evaluations, we provide the LLM judge with the ground truth human-annotated warning labels for the specific conversational turn being evaluated. Therefore, as demonstrated through the evaluation prompts in Appendix A.10, the LLM judge *does not perform the warning classification task*, but rather a much simpler semantic matching task to identify whether or not the provided conversational turn contains the correct warnings and/or determine which of the two provided responses better reflects the ground truth warning labels. We agree that this is an important detail about the evaluation, and have clarified this in Section 4.2 (line 371) of the updated version.
>
> We would also like to note that the LLM judges are *well-calibrated*. Specifically, as stated on line 410, the Pearson correlation between LLM judge ratings and human-provided ratings is 0.74, and the agreement rate is 90%. Additionally, from Table 4, the human-oracle rewritten responses are almost always preferred to the original GPT-4o response, indicating the GPT-4o judge is not simply preferring its own response. Also, even the best model after fine-tuning only achieves a roughly 50% win-rate against the human-oracle, indicating that the fine-tuned model responses are nearly indistinguishable from the oracle responses, which is the upper-bound of possible improvement. We have highlighted these points in Section 5.3 of the revised paper.
>
> > The evaluation framework fails to assess the model's ability to perceive physical danger from visual context, which is a critical component of the very problem the paper aims to solve.
>
> Thank you for pointing this out. We explored incorporating images alongside text in our experiments using state-of-the-art multi-modal methods. However, we did not find them particularly helpful. We hypothesize that the images provided in instructional documents are not crucial for generating safety warnings because oftentimes the images simply provide additional context for the textual instructions. However, we do not rule out the possibility of future work showing some way to use the images to improve warning generation performance. By including these images in the released dataset, we hope future research will further investigate how such visual demonstrations might enhance physical safety awareness. We have added this discussion to the Appendices in the Limitations section.
>
> > The Dataset Name "SafetyChat" is a Significant Overclaim: The name implies a general-purpose safety model, whereas the work is narrowly focused only on procedural physical safety for automotive and electronics repair.
>
> We acknowledge that the name SafetyChat could cause misconceptions about the generality of our benchmark beyond physical safety. In the revised version of the paper, we have changed the name of the dataset to PhySafe and have accordingly updated the title.
>
> > How do the authors view the generalization capabilities of models trained on SAFETYCHAT? … Has the model learned specific textual patterns for "jack safety" and "battery safety," or has it grasped a more general concept of "physical harm avoidance"?
>
> Thank you for raising this point. We experimented with the cross-domain settings where the model is finetuned on the Autorepair/WikiHow Cars subsets and tested on Autorepair/Ford (5/10 unseen classes) and Electronics/iFixit subsets (6/8 unseen classes). We add a new result table to the appendix. The finetuned models perform significantly better than out-of-the-box models on the new domains and unseen warning classes, which suggests that our finetuning is more than textual pattern matching. However, there are some other classes where the model struggles, potentially suggesting a negative transfer where the model over-predicts the unseen classes if their names are similar to seen classes, especially for closer domains. We leave it as future work to explore more efficient generalization.

---

> ### Author Response · Authors · 2025-12-02
> **Updated tables**
>
> Transfer learning experiments
>
> # Autorepair/Ford
>
> | Method - Binary F1        | SS   | CD   | JS   | ES   | UTC  | WPE  | MH   | RWD  | CE   | Di   | Average |
> |---------------------------|------|------|------|------|------|------|------|------|------|------|---------|
> | Random                    | 0.11 | 0.11 | 0.03 | 0.06 | 0.04 | 0.15 | 0.12 | 0.03 | 0.08 | 0.08 | 0.13 |
> | No Warning                | 0    | 0    | 0    | 0    | 0    | 0    | 0    | 0    | 0    | 0    | 0    |
> | GPT-4.1-mini-0s           | 0.11 | 0.19 | 0.11 | 0.40 | 0.02 | 0.16 | 0.16 | 0    | **0.32** | 0.37 | 0.18 |
> | GPT-4.1-mini-4s           | 0.30 | 0.62 | 0    | 0.33 | 0.04 | 0.21 | 0.30 | 0    | 0.27 | 0.48 | 0.26 |
> | Llama-3.1-8B-0s           | 0.16 | 0.22 | 0.11 | 0.08 | 0.05 | 0.28 | 0.03 | 0    | 0    | 0.30 | 0.12 |
> | Llama-3.1-8B-8s           | 0.19 | 0.22 | 0.10 | 0.15 | 0.05 | 0.27 | 0.14 | 0.18 | 0    | 0.36 | 0.17 |
> | Llama-3.1-8B-CoT-0s       | 0.30 | 0.28 | 0.05 | 0.10 | 0.04 | 0.19 | 0.08 | 0    | 0    | 0.32 | 0.14 |
> | Llama-3.1-8B-CoT-8s       | 0.27 | 0.31 | 0.10 | 0.10 | 0.07 | 0.31 | 0.16 | 0.07 | 0    | 0.32 | 0.17 |
> | Llama-3.1-8B-RAG-0s       | 0.16 | 0.22 | 0.11 | 0.08 | 0.05 | 0.28 | 0.03 | 0    | 0    | 0.30 | 0.12 |
> | Llama-3.1-8B-RAG-8s       | 0.19 | 0.22 | 0.10 | 0.15 | 0.05 | 0.27 | 0.14 | 0.18 | 0    | 0.36 | 0.17 |
> | **Llama-3.1-8B-SFT**      | **0.63** | **0.71** | 0.50 | 0.36 | **0.50** | **0.50** | **0.52** | **0.80** | 0.29 | 0.47 | **0.53** |
> | **Llama-3.1-8B-SFT(transferred from WikiHow)** | 0.63 | 0.57 | 0.40 | 0.36 | 0    | 0.57 | 0.29 | 0    | 0.10 | 0.52 | 0.34 |
> | Qwen-2.5-7B-0s            | 0.24 | 0.48 | 0    | 0.16 | 0.11 | 0.33 | 0.09 | 0.29 | 0    | 0.32 | 0.20 |
> | Qwen-2.5-7B-8s            | 0.24 | 0.48 | 0    | 0.24 | 0.08 | 0.22 | 0.17 | 0    | 0    | 0.25 | 0.17 |
> | Qwen-2.5-7B-CoT-0s        | 0.37 | 0.56 | 0    | 0.28 | 0.13 | 0.42 | 0.03 | 0    | 0    | 0.38 | 0.22 |
> | Qwen-2.5-7B-CoT-8s        | 0.35 | 0.53 | 0    | 0.26 | 0.06 | 0.40 | 0.25 | 0    | 0    | 0.32 | 0.22 |
> | Qwen-2.5-7B-RAG-0s        | 0.24 | 0.48 | 0    | 0.16 | 0.11 | 0.33 | 0.09 | 0.29 | 0    | 0.32 | 0.20 |
> | Qwen-2.5-7B-RAG-8s        | 0.24 | 0.48 | 0    | 0.24 | 0.08 | 0.22 | 0.17 | 0    | 0    | 0.25 | 0.17 |
> | **Qwen-2.5-7B-SFT**       | 0.62 | 0.65 | **0.55** | **0.37** | **0.50** | 0.47 | 0.37 | 0.67 | 0.18 | **0.50** | 0.49 |
>
> # Electronics/iFixit
> | Method - Binary F1        | ES   | HFS  | SWE  | WPE  | TU   | SS   | HMT  | Di   | Average |
> |---------------------------|------|------|------|------|------|------|------|------|---------|
> | Random                    | 0.22 | 0.02 | 0.06 | 0.05 | 0.08 | 0.03 | 0.02 | 0.12 | 0.07 |
> | No Warning                | 0    | 0    | 0    | 0    | 0    | 0    | 0    | 0    | 0    |
> | GPT-4o-0-shot             | 0.24 | 0.33 | 0.24 | 0.42 | **0.23** | 0.57 | 0 | 0.11 | 0.27 |
> | GPT-4o-8-shot             | 0.39 | 0.57 | 0.20 | 0.15 | 0.15 | 0.57 | 0 | 0.11 | 0.27 |
> | Llama-3.1-8B-0-shot       | 0.33 | 0.12 | 0    | 0.08 | 0.08 | 0.27 | 0 | 0 | 0.11 |
> | Llama-3.1-8B-8-shot       | 0.32 | 0.16 | 0.07 | 0.13 | 0.08 | 0.15 | 0 | 0 | 0.11 |
> | Llama-3.1-8B-CoT-0-shot   | 0.32 | 0.07 | 0.02 | 0.13 | 0.09 | 0.33 | 0 | 0.09 | 0.13 |
> | Llama-3.1-8B-CoT-8-shot   | 0.31 | 0.16 | 0.05 | 0.12 | 0.13 | 0.33 | 0 | **0.16** | 0.16 |
> | Llama-3.1-8B-RAG-0-shot   | 0.33 | 0.12 | 0    | 0.08 | 0.08 | 0.27 | 0 | 0 | 0.11 |
> | Llama-3.1-8B-RAG-8-shot   | 0.32 | 0.16 | 0.07 | 0.13 | 0.08 | 0.15 | 0 | 0 | 0.11 |
> | **Llama-3.1-8B-SFT**      | **0.56** | 0 | **0.31** | **0.67** | 0.15 | **0.80** | 0 | 0.11 | **0.32** |
> | **Llama-3.1-8B-SFT(Transferred from WikiHow)**| 0.36 | 0.50 | 0 | 0.50 | 0 | 0.67 | 0 | 0 | 0.25 |
> | Qwen-2.5-7B-0-shot        | 0.44 | **0.67** | 0.07 | 0.20 | 0.04 | 0.57 | 0 | 0 | 0.25 |
> | Qwen-2.5-7B-8-shot        | 0.43 | 0.57 | 0.16 | 0.22 | 0.12 | 0.67 | 0 | 0.11 | 0.29 |
> | Qwen-2.5-7B-CoT-0-shot    | 0.38 | **0.67** | 0.11 | 0.11 | 0.05 | 0.67 | 0 | 0.11 | 0.26 |
> | Qwen-2.5-7B-CoT-8-shot    | 0.38 | 0.33 | 0.08 | 0.09 | 0 | 0.57 | 0 | 0 | 0.18 |
> | Qwen-2.5-7B-RAG-0-shot    | 0.44 | **0.67** | 0.07 | 0.20 | 0.04 | 0.57 | 0 | 0 | 0.25 |
> | Qwen-2.5-7B-RAG-8-shot    | 0.43 | 0.57 | 0.16 | 0.22 | 0.12 | 0.67 | 0 | 0.11 | 0.29 |
> | **Qwen-2.5-7B-SFT**       | 0.52 | 0.50 | 0.11 | 0.44 | 0.22 | 0.67 | 0 | 0 | 0.31 |

---

### Author Response · Authors · 2025-12-01
**Summary of changes**

Dear reviewers,

Thank you for taking the time to review our paper. We have read every comment and found them very useful. We add experiments based on your suggestions, which takes us some time.

We would like to summarize the changes being made to the paper during this discussion period to save your time.
1. Additional baselines.

We added several new baselines, including chain-of-thought (CoT) reasoning and retrieval-augmented generation (RAG). In our experiments, both approaches achieved improvements over the direct prompting baseline, although domain-specific finetuning remains more effective overall.

2. Cross-domain generalization experiments.

We conducted cross-domain evaluations by finetuning the model on the autorepair/Wikihow subset and testing on two unseen target domains: autorepair/Ford (5/10 unseen classes) and electronics/iFixit (6/8 unseen classes). These experiments assess how well models trained on our dataset generalize to new domains. We found that the finetuned model substantially outperforms out-of-the-box LLMs on these unseen domains, though it remains slightly behind domain-specific finetuning. Due to space constraints, a deeper investigation of transfer learning across our dataset is left for future work.

3. Anonymous GitHub repository.

We created an anonymized GitHub repository containing our code, dataset processing scripts, and instructions for reproducing our experiments. This repository is included in the updated submission to facilitate reproducibility while preserving anonymity.

https://anonymous.4open.science/r/SafetyChat-FAD6

4. New results

All the newly added methods were tested and added to the result tables 3, 6, 7, and 8. Below is an example. More could be found in the appendices
| Warning Classes | BS-P | BS-R | SS-P | SS-R | CD-P | CD-R | JS-P | JS-R | WPE-P | WPE-R | Fo-P | Fo-R | Fl-P | Fl-R | Di-P | Di-R | All-P | All-R | All-F |
|-----------------|------|------|------|------|------|------|------|------|--------|--------|------|------|------|------|------|------|-------|-------|-------|
| Random          | 8    | 48   | 6    | 52   | 5    | 66   | 3    | 56   | 5      | 50     | 4    | 57   | 9    | 43   | 2    | 55   | 5     | 53    | 10    |
| No Warning      | 0    | 0    | 0    | 0    | 0    | 0    | 0    | 0    | 0      | 0      | 0    | 0    | 0    | 0    | 0    | 0    | 0     | 0     | 0     |
| Human           | 96   | 96   | 97   | 96   | 98   | 95   | 98   | 100  | 88     | 78     | 80   | 91   | 97   | 94   | 100  | 96   | 94    | 93    | 93    |
| GPT-4o-0-shot   | 57   | 79   | 41   | 17   | 69   | 38   | 60   | 83   | 20     | 44     | 100  | 7    | 46   | 26   | 70   | 64   | 58    | 45    | 44    |
| GPT-4o-8-shot   | 64   | 86   | 43   | 45   | 67   | 41   | 76   | 89   | 24     | 64     | 30   | 79   | 33   | 43   | 56   | 91   | 49    | 67    | 54    |
| Llama-3.1-8B-0-shot | 21 | 90 | 19 | 52 | 17 | 69 | 12 | 94 | 12 | 25 | 5 | 43 | 18 | 48 | 14 | 82 | 15 | 63 | 23 |
| Llama-3.1-8B-8-shot | 20 | 69 | 17 | 52 | 15 | 62 | 11 | 83 | 10 | 22 | 8 | 64 | 12 | 29 | 12 | 64 | 13 | 56 | 21 |
| Llama-3.1-8B-CoT-0-shot | 28 | 83 | 18 | 60 | 24 | 62 | 12 | 94 | 14 | 33 | 10 | 71 | 13 | 40 | 10 | 55 | 16 | 62 | 25 |
| Llama-3.1-8B-CoT-8-shot | 27 | 86 | 19 | 62 | 16 | 52 | 11 | 83 | 12 | 28 | 7 | 50 | 14 | 40 | 12 | 73 | 15 | 59 | 23 |
| Llama-3.1-8B-RAG-0-shot | 22 | 59 | 17 | 45 | 24 | 34 | 11 | 61 | 13 | 33 | 8 | 71 | 13 | 36 | 11 | 55 | 15 | 49 | 22 |
| Llama-3.1-8B-RAG-8-shot | 18 | 52 | 22 | 67 | 17 | 28 | 14 | 67 | 11 | 28 | 5 | 50 | 12 | 33 | 12 | 55 | 14 | 47 | 21 |
| **Llama-3.1-8B-SFT** | 59 | 45 | 89 | 40 | 87 | 45 | 94 | 89 | 80 | 33 | 55 | 43 | 71 | 29 | 100 | 73 | 79 | 50 | **60** |
| Qwen-2.5-7B-0-shot | 51 | 76 | 23 | 40 | 33 | 45 | 42 | 83 | 32 | 28 | 19 | 64 | 32 | 29 | 33 | 18 | 33 | 48 | 37 |
| Qwen-2.5-7B-8-shot | 60 | 86 | 24 | 40 | 37 | 45 | 38 | 83 | 31 | 22 | 18 | 50 | 33 | 31 | 60 | 27 | 38 | 48 | 39 |
| Qwen-2.5-7B-CoT-0-shot | 67 | 76 | 27 | 38 | 52 | 55 | 44 | 78 | 41 | 33 | 21 | 57 | 41 | 40 | 50 | 45 | 43 | 43 | 46 |
| Qwen-2.5-7B-CoT-8-shot | 66 | 72 | 28 | 40 | 48 | 48 | 48 | 89 | 38 | 31 | 19 | 50 | 39 | 31 | 43 | 27 | 41 | 49 | 43 |
| Qwen-2.5-7B-RAG-0-shot | 57 | 79 | 25 | 45 | 50 | 34 | 33 | 83 | 27 | 28 | 13 | 36 | 28 | 31 | 40 | 18 | 34 | 44 | 36 |
| Qwen-2.5-7B-RAG-8-shot | 57 | 83 | 20 | 40 | 30 | 28 | 41 | 89 | 28 | 25 | 13 | 43 | 19 | 21 | 30 | 27 | 30 | 45 | 34 |
| **Qwen-2.5-7B-SFT** | **71** | 34 | 81 | 52 | 69 | 62 | 94 | 94 | 72 | 36 | 78 | 50 | 62 | 24 | 78 | 64 | 76 | 52 | **60** |

5. Dataset name change
As suggested by one reviewer, our dataset name “SafetyChat” didn’t specify the physical safety focus of the dataset, as opposed to general safety. We acknowledge that the name SafetyChat could cause misconceptions about the generality of our benchmark beyond physical safety. In the revised version of the paper, we have changed the name of the dataset to PhySafe and have accordingly updated the title.

---

### Meta-Review · Area_Chair_Wx2h · 2025-12-28

**Summary:**

This paper addresses an important and overlooked problem: the inability of Large Language Models (LLMs) to generate context-aware safety warnings when acting as instructional assistants for complex tasks with physical risks (e.g., automotive or electronics repair). The authors point out that even advanced models like GPT-4o often fail to provide critical physical safety instructions.

**Reviewer Concerns:**

* The authors only provided LLM–human correlation statistics (Pearson 0.74, 90% agreement) based on a small calibration study, not a standalone expert evaluation.

* Although clarified that judges perform semantic matching rather than hazard detection, the core concern remains: evaluation still depends almost entirely on LLM judges.

* Reviewers explicitly asked for out-of-domain transfer (e.g., kitchen tasks, home improvement, general DIY). Added “cross-domain” experiments remain within closely related repair domains (automotive ↔ electronics), which does not resolve concerns about broader physical-safety generalization.

* The authors report over-prediction and confusion among unseen classes with similar names. No methodological or analytical follow-up is provided to explain, diagnose, or mitigate this issue

* Despite acknowledging the importance of images in repair tasks, the authors conclude (without strong evidence) that images are “not particularly helpful.” No quantitative multimodal results are shown; the claim rests on informal exploration.

* While training and reannotation procedures are described, no quantitative IAA (e.g., Cohen’s κ, Krippendorff’s α) is provided.

* Reviewer concern about preference mismatch (human-written vs. GPT-4o-generated negatives) is not addressed with stability analysis, ablations, or convergence diagnostics.

**Reviewer Scores:**

remain unchanged

---

### Decision · Program_Chairs · 2026-01-26

Reject